# RST-YOLOv8: An Improved Chip Surface Defect Detection Model Based on YOLOv8

**DOI:** 10.3390/s25133859

**Published:** 2025-06-21

**Authors:** Wenjie Tang, Yangjun Deng, Xu Luo

**Affiliations:** 1College of Information and Intelligence, Hunan Agricultural University, Changsha 410128, China; tangwenjie66@stu.hunau.edu.cn (W.T.); dengyangjun@hunau.edu.cn (Y.D.); 2Hunan Provincial Engineering and Technology Research Center for Rural and Agricultural Informatization, Hunan Agricultural University, Changsha 410128, China; 3Yuelushan Laboratory, Changsha 410128, China

**Keywords:** chip surface defect detection, RST-YOLOv8, feature representation, small object detection

## Abstract

Surface defect detection in chips is crucial for ensuring product quality and reliability. This paper addresses the challenge of low identification accuracy in chip surface defect detection, which arises from the similarity of defect characteristics, small sizes, and significant scale differences. We propose an enhanced chip surface defect detection algorithm based on an improved version of YOLOv8, termed RST-YOLOv8. This study introduces the C2f_RVB module, which incorporates RepViTBlock technology. This integration effectively optimizes feature representation capabilities while significantly reducing the model’s parameter count. By enhancing the expressive power of deep features, we achieve a marked improvement in the identification accuracy of small defect targets. Additionally, we employ the SimAM attention mechanism, enabling the model to learn three-dimensional channel information, thereby strengthening its perception of defect characteristics. To address the issues of missed detections and false detections of small targets in chip surface defect detection, we designed a task-aligned dynamic detection head (TADDH) to facilitate interaction between the localization and classification detection heads. This design improves the accuracy of small target detection. Experimental evaluations on the PCB_DATASET indicate that our model improved the mAP@0.5 by 10.3%. Furthermore, significant progress was achieved in experiments on the chip surface defect dataset, where mAP@0.5 increased by 5.4%. Simultaneously, the model demonstrated significant advantages in terms of computational complexity, as both the number of parameters and GFLOPs were effectively controlled. This showcases the model’s balance between high precision and a lightweight design. The experimental results show that the RST-YOLOv8 model has a significant advantage in detection accuracy for chip surface defects compared to other models. It not only enhances detection accuracy but also achieves an optimal balance between computational resource consumption and real-time performance, providing an ideal technical pathway for chip surface defect detection tasks.

## 1. Introduction

With the rapid advancement of integrated circuit (IC) technology, the manufacturing processes for chips have become increasingly complex, making chip surface defect detection a crucial step in ensuring both quality and functionality [1]. Traditional manual inspection methods are inefficient and prone to human error, which has prompted the development of automated detection technologies as a vital solution to this challenge. Object detection technology, particularly deep learning-based methods, has shown significant promise in chip surface defect detection due to its efficient feature extraction and precise object localization capabilities.

Since the introduction of deep learning and deep convolutional neural networks, these technologies have garnered widespread attention and have been increasingly applied to industrial inspection, yielding significant results [2]. These applications can be broadly categorized into one-stage and two-stage models. One-stage models are characterized by fast detection speeds but lower accuracy, with representative algorithms including the YOLO [3,4,5,6,7,8,9] series and SSD [10]. In contrast, two-stage models generally achieve higher detection accuracy at the expense of speed, exemplified by region-based convolutional neural networks such as R-CNN [11], faster R-CNN [12], and mask R-CNN [13]. Among various object detection methods, the YOLO series is particularly notable for its real-time performance and high accuracy, making it widely used across different object detection tasks. The YOLO series, despite its success, still encounters significant challenges in detecting small targets, particularly in chip surface defect detection, where even subtle defects are often difficult to identify accurately. In response, a growing number of researchers are turning to deep convolutional neural networks for the detection and classification of chip surface defects. For example, Li et al. [14] incorporated the SSD network as a meta-structure, combining it with the MobileNet [15] convolutional network to develop MobileNet-SSD. This approach optimizes the SSD architecture, adjusting the network structure and parameters to simplify the detection model. Meanwhile, Qian et al. [16] employed ShuffleNetv2 [17] to streamline feature extraction, creating a lightweight feature pyramid network (LFPN) to enhance fusion efficiency. Additionally, they introduced the adaptive receptive field feature extraction (ARFFE) module to tackle multi-scale challenges. Zheng et al. [18] developed an improved real-time detection algorithm using YOLOv4, which integrates additional positional and semantic information to enhance the accuracy of multi-scale object detection and reduce false positives. Liu et al. [19] improved the effectiveness of bounding box regression for small and irregularly shaped objects by replacing the CIOU loss function with the EIOU loss function, thereby enhancing the measurement of similarity between these objects. Zhai et al. [20] designed the C3GhostNetV2 module and combined it with the CA [21] (coordinate attention) mechanism to improve the detection capability of YOLOv7 for small targets while reducing the model’s complexity. Diao et al. [22] incorporated the adaptive feature pyramid network (AFPN) into YOLOv8 to replace the original structure, enhancing the fusion capabilities of different feature layers. Additionally, they introduced the multi-layer interaction module (MLIM) based on self-attention mechanisms to strengthen the learning of both shallow and deep features, thereby improving the model’s overall learning capacity and contextual information aggregation. Lin et al. [23] proposed the feature pyramid network (FPN), which leverages the inherent pyramid hierarchy of deep convolutional networks to construct feature pyramids at a lower cost, thereby improving the performance of multi-scale object detection. Jiao et al. [24] proposed a scene context-aware salient object detection method that explicitly leverages semantic scene context information to enhance the performance of salient object detection in complex environments. Feng et al. [25] conducted a comprehensive review of small object detection techniques based on deep learning. They analyzed the challenges and solutions associated with small object detection from four perspectives: enhancing the resolution of input features, implementing scale-aware training, integrating contextual information, and employing data augmentation. Additionally, they proposed directions for future research. Li et al. [26] proposed a high-precision detection method for surface defects on the inner walls of cylindrical parts, utilizing an optimized YOLOv8 algorithm and random cropping augmentation (RCA). By incorporating a small target detection layer, the detection accuracy was significantly improved. Liu et al. [27] proposed a lightweight, real-time surface defect detection method based on YOLOv4. By enhancing the network structure and employing data augmentation strategies, they significantly improved detection speed while maintaining accuracy and reducing the model’s size. Ling et al. [28] proposed a dense detection method for PCB components based on an improved version of YOLOv8. By incorporating the C2Focal module, Ghost convolution, and Sig-IoU loss function, this approach achieves rapid and accurate detection of densely packed PCB components.

Despite these advancements, the localization and defect detection of chips continues to face significant challenges. For instance, when dealing with complex background environments, most methods struggle to effectively extract the features of small and blurred targets, resulting in a high rate of missed and false detections of chip surface defects. Consequently, optimizing the YOLOv8 model to improve its ability to detect small-target defects has become the central focus of this study. This paper comprehensively considers the aforementioned defects while also taking into account the real-time performance of detection. Based on the YOLOv8 model, an improved method is proposed. The main improvements of this paper are as follows:

(1) To address the challenges of detecting small defects in low-resolution images, this paper enhances the C2f module by integrating the RepViTBlock, which improves the network’s ability to process features and refine feature representation. The RepViTBlock enhances multi-scale feature extraction, helping the model detect minor defects more effectively.

(2) The SimAM attention mechanism, originally proposed by Yang et al. in 2021 [29], is introduced to further boost the model’s capacity to detect small defects in complex backgrounds. By utilizing three-dimensional attention weights, SimAM enhances feature discrimination and reduces background noise, improving detection accuracy.

(3) To handle variations in defect sizes and their similarities, the paper proposes a task-aligned dynamic detection head (TADDH), which aligns object classification and localization tasks through task interaction features. This approach enhances the model’s ability to detect defects of varying sizes and similar characteristics, improving both precision and robustness.

The structure of this paper is organized as follows: Section 2 reviews the relevant research and background developments of the YOLOv8 model. Section 3 provides a comprehensive introduction to the proposed RST-YOLOv8 model, detailing its architecture and key enhancements. Section 4 presents the experimental setup and analyzes the results, comparing the performance of different models. Finally, Section 5 concludes the paper by summarizing its main contributions and discussing potential directions for future research.

## 2. Overview of YOLOv8

The YOLOv8 architecture is designed with five distinct model sizes, n, s, l, m, and x, to accommodate various application scenarios and requirements. As the network depth increases and the number of parameters expands, the detection accuracy correspondingly improves. In this study, we selected the YOLOv8n model, which features fewer parameters and lower computational complexity, effectively controlling the model size while ensuring the algorithm’s real-time performance, thereby fully meeting our practical needs.

The network structure of YOLOv8n comprises three main components: the backbone, neck, and head. The backbone utilizes the Darknet-53 framework and includes Conv, C2f, and SPPF (spatial pyramid pooling fusion) structures. The C2f module, inherited from the ELAN structure in YOLOv7, demonstrates fewer parameters and superior feature extraction capabilities compared to the C3 module in YOLOv5. The SPPF module is responsible for standardizing the vector size of feature maps at different scales. The neck employs multi-scale feature fusion technology, integrating concepts from PAN [30] (path aggregation networks) and FPN [31] (feature pyramid networks) to merge features from various stages of the backbone network, thereby enhancing feature representation capabilities. The head adopts the decoupling concept from YOLOX [32], transitioning from the anchor-based coupled head of YOLOv5 to an anchor-free decoupled head, which separates category features from location features, thus improving detection efficiency. Regarding loss calculation, the classification task utilizes BCE Loss, while the regression task employs distribution focal loss and CIOU loss functions.

## 3. Methodology

### 3.1. RST-YOLOv8

The structure of the RST-YOLOv8 network model is illustrated in Figure 1. First, the network replaces the original C2f module with C2f_RVB, which not only reduces the number of model parameters and computational complexity but also enhances detection accuracy and processing speed. Additionally, a SimAM attention mechanism is introduced after spatial pyramid pooling, enabling collaborative attention between the spatial and channel parameters of the image, thereby further improving the network’s detection accuracy. Finally, to address the challenges of detecting small target defects in object detection, a dynamic task detection head has been designed to enhance the model’s capability to detect small target defects.

### 3.2. RepViTBlock

In the task of object detection, convolutional neural networks (CNNs) are widely utilized due to their exceptional feature extraction capabilities. However, as model architectures become increasingly complex, these networks often demand significant computing resources and storage space. To address this challenge, Wang et al. [33] proposed a novel network structure unit called the RepViTBlock. The RepViTBlock integrates the design philosophy of lightweight visual transformers (ViTs) with the practical advantages of convolutional neural networks, aiming to enhance model efficiency while reducing the parameter count. The structure of the RepViTBlock is illustrated in Figure 2.

In the design of the RepViTBlock, the data flow first passes through the token mixer, which utilizes depthwise separable convolution (RepVGGDW). As illustrated in Figure 3, RepVGGDW consists of two convolutional layers: depthwise convolution and 1 × 1 grouped convolution. The outputs of these two convolutions are summed and then fused with the original input to create a residual connection, followed by normalization through a batch normalization (BN) layer before the output. This stage primarily processes features from local or neighboring regions, enhancing the model’s ability to capture spatial details. The 1 × 1 grouped convolution reduces computational load by dividing the convolution operation into multiple groups, with each group processing only its assigned input channels rather than all input channels. The RepViTBlock integrates depthwise separable convolution and 1 × 1 grouped convolution, utilizing structural reparameterization techniques to decrease both the number of parameters and computational requirements. This approach enables the module to maintain efficient feature processing while keeping computational costs low.

Subsequently, the features are transferred to the channel_mixer, where inter-channel feature fusion is conducted through two 1 × 1 convolutional layers. This component not only enhances the expressive power of the features but also facilitates dimensional transformation and feature compression, thereby assisting the model in integrating information from various channels and achieving comprehensive global feature integration.

### 3.3. C2f_RVB

The C2f module in YOLOv8 employs multiple bottleneck structures for feature fusion and information propagation. The bottleneck module incorporates feature partitioning and residual connections. However, the repeated use of the bottleneck module for feature processing increases both the complexity and computational burden of the network. To address this issue, the C2f_RVB enhances the network’s feature processing capability by utilizing the RepViTBlock to replace the bottleneck in the C2f module. This design allows C2f_RVB to integrate local and global information more effectively, thereby improving the richness and accuracy of feature representation. Additionally, the implementation of the RepViTBlock optimizes computational efficiency by reducing the number of parameters and computational requirements through the use of depthwise separable convolutions. The structure of C2f_RVB is illustrated in Figure 4.

### 3.4. SimAM Attention Mechanism

The attention mechanism can generally be categorized into spatial attention mechanisms and channel attention mechanisms. The spatial attention mechanism evaluates each pixel in the image and computes a weighted average of the scores to emphasize areas of greater focus. In contrast, the channel attention mechanism integrates and associates information across different channels, thereby enhancing the model’s representational capability. Compared to traditional spatial and channel attention mechanisms, the SimAM attention mechanism introduces a unified weight attention module that can directly compute three-dimensional attention weights. The integration process of the three-dimensional attention mechanism is illustrated in Figure 5. SimAM is a non-parametric attention mechanism that enhances the responses of important feature channels by calculating the inter-channel relationships of input features and making dynamic weight adjustments. The core idea is to improve the model’s representational capacity by adaptively learning the weights of feature channels without increasing the number of additional model parameters.

The SimAM attention mechanism integrates concepts from visual neuroscience, where each neuron possesses varying levels of importance. Active neurons can suppress the activity of adjacent neurons, making it essential to prioritize those neurons that exhibit a significant spatial suppression effect. Consequently, the following energy function is defined for each neuron:(1)$$e_{t}\left(w_{t},b_{t},y,x_{i}\right)={\left(y_{t}-\hat{t}\right)}^{2}+\frac{1}{M-1}\sum_{i=1}^{M-1}{\left(y_{0}-{\hat{x}}_{i}\right)}^{2}.$$

In the formula, $e_{t}$ represents the energy function of the target neuron *t*. The variables $w_{t}$ and $b_{t}$ denote the weights and biases of the linear transformation, while *y* indicates the output of the neuron. Specifically, $y_{t}$ refers to the output of the target neuron, $y_{0}$ represents the outputs of other neurons, and $x_{i}$ refers to the remaining neurons. Additionally, *M* denotes the number of neurons within the channel. The terms $\hat{t}$ and ${\hat{x}}_{i}$ are derived from *t* and $x_{i}$ through linear transformations, as illustrated in Equation (Equation 2).(2)$$\left\{\begin{array}{c} {\hat{x}}_{i}=x_{i}w_{t}+b_{t}\\ \hat{t}=tw_{t}+b_{t} \end{array}.\right.$$

Finding the minimum value of the energy function allows us to identify the linear separability of the target function *t* with respect to the other neurons within the same channel. Typically, $y_{t}$ and $y_{0}$ are represented using binary labels, namely −1 and 1. Additionally, regularization is incorporated into the equation, and the final energy function is expressed as follows:(3)$$e_{t1}\left(w_{t},b_{t},y,x_{i}\right)=\frac{1}{M-1}\sum_{i=1}^{M-1}{\left(-1-\left(w_{t}x_{i}+b_{t}\right)\right)}^{2},$$(4)$$e_{t2}\left(w_{t},b_{t},y,x_{i}\right)={\left(1-\left(w_{t}t+b_{t}\right)\right)}^{2}+\lambda w_{t}^{2},$$(5)$$e_{t}\left(w_{t},b_{t},y,x_{i}\right)=e_{t1}\left(w_{t},b_{t},y,x_{i}\right)+e_{t2}\left(w_{t},b_{t},y,x_{i}\right).$$

A rapid closed-form solution to $w_{t}$ and $b_{t}$ is derived from Equation (Equation 6).(6)$$\left\{\begin{array}{c} w_{t}=-\frac{2\left(t-\mu_{t}\right)}{{\left(t-\mu_{t}\right)}^{2}+2\sigma_{t}^{2}+2\lambda }\\ b_{t}=-\frac{1}{2}\left(t+\mu_{t}\right)w_{t} \end{array}.\right.$$

In the equation, the values $\mu_{t}=\frac{1}{M-1}\sum_{i=1}^{M-1}x_{i}$ and $\sigma_{t}^{2}=\frac{1}{M-1}\sum_{i}^{M-1}{\left(x_{i}-\mu_{t}\right)}^{2}$ represent the mean and variance calculated across all neurons in the channel, excluding *t*. Since the solution presented in Equation (Equation 5) is derived from a single channel, it is assumed that all pixels within the channel follow the same distribution. Based on this assumption, the mean and variance are calculated for all neurons and subsequently reapplied to all neurons within the channel. Consequently, the minimum energy can be calculated using Equation (Equation 7):(7)$$e_{t}^{*}=\frac{4\left({\hat{\sigma }}^{2}+\lambda \right)}{{\left(t-\hat{\mu }\right)}^{2}+2{\hat{\sigma }}^{2}+2\lambda },$$
where $\mu_{t}=\frac{1}{M}\sum_{i=1}^{M}x_{i}$ and $\sigma_{t}^{2}=\frac{1}{M}\sum_{i}^{M}{\left(x_{i}-\mu_{t}\right)}^{2}$ represent the mean and variance of all neurons, respectively. Equation (Equation 7) demonstrates that as the energy decreases, the distinction between a neuron and its surrounding neurons increases, resulting in a higher weight for the neuron. Equation (Equation 8) corresponds to the refinement phase of the entire module:(8)$$\tilde{X}=\mathrm{sigmoid}\left(\frac{1}{E}\right)⊙X.$$

In this context, the sigmoid function is employed to limit excessively large values in $\frac{1}{E}$, where *E* represents the total sum of all $e_{t}^{*}$ across both the channel and spatial dimensions. Here, *X* denotes the feature map of the image input, while $\tilde{X}$ represents the processed result obtained after applying a weighted transformation to the input features.

### 3.5. TADDH

Due to issues such as misdetections and high miss rates in detecting small targets in chip surface defects using the traditional YOLOv8 algorithm, this paper proposes the addition of a task-aligned dynamic detection head (TADDH) to YOLOv8 to address these problems. The structural diagram is shown in Figure 6. Single-stage object detection typically optimizes both classification and localization tasks, often adopting a dual-branch head design. However, this approach may lead to spatial inconsistency between predictions. Additionally, such a dual-branch design may result in a lack of interaction between the two tasks, potentially causing inconsistent predictions during task execution. Inspired by the task-aligned one-stage object detection (TOOD) concept proposed by Feng et al. [34], this paper introduces a task-aligned single-stage object detection head (TADDH). The TADDH aims to more accurately align the two tasks by implementing a new head structure and an alignment-oriented learning approach. Specifically, TADDH dynamically adjusts the importance of features across different tasks by introducing a hierarchical attention mechanism and enhances the synergy between classification and localization tasks through the design of a novel gradient alignment strategy.

In Figure 6, task decomposition refers to the process of breaking down the features obtained from concatenation. Objective classification and localization are performed on the computed task interaction features, allowing the two tasks to effectively perceive one another’s states. However, the design of a single branch inevitably introduces feature conflicts between the tasks. These conflicts arise from the differing objectives of classification and localization, which focus on distinct types of features, such as varying levels and receptive fields. To address this, the paper proposes a layer attention mechanism that dynamically computes task-specific features at different levels, facilitating task decomposition, as illustrated in Figure 7.

For classification and localization, calculate their task-specific features separately:(9)$$X_{k}^{\mathrm{task}\;}=\omega_{k}\cdot X_{k}^{\mathrm{inter}}\;,\forall k\in \left\{1,2,\ldots ,N\right\}.$$

Among them, $\omega_{k}$ represents the k-th element of the learned layer attention weights $\omega \in R^{N}$. The weights ω are computed based on the cross-layer task interaction features and are capable of capturing the interaction relationships between layers:(10)$$\omega =\sigma \left(fc_{2}\left(\sigma \left(fc_{1}\left(x^{\mathrm{inter}\;}\right)\right)\right)\right)\sigma .$$

Among them, fc1 and fc2 are two fully connected layers, and the activation function used is the sigmoid function. The variable $x^{inter}$ is obtained by applying average pooling to $X^{inter}$, while $X^{inter}$ itself is derived by concatenating $X_{k}^{inter}$.(11)$$Z^{task}=conv_{2}\left(\delta \left(conv_{1}\left(X^{task}\right)\right)\right).$$

Among these, $x^{task}$ represents a cascade feature of $X^{task}$, while conv1 is a 1×1 convolution applied for dimensionality reduction. Additionally, $Z^{task}$ denotes a dense classification score $P\in R^{H\times W\times 80}$, transformed using the sigmoid function, or a predicted bounding box size $B\in R^{H\times W\times 4}$ in the regression task. This configuration facilitates the alignment of both classification and localization tasks.

The incorporation of this method significantly enhances the localization and classification of chip surface defect detection, particularly yielding favorable results in the detection of small targets within this domain.

## 4. Experiments

### 4.1. Datasets

The PCB defect detection dataset (PKU-Market-PCB) [35] is derived from a publicly available collection released by the Intelligent Robotics Open Lab at Peking University. This dataset includes six primary defect categories: missing hole, mouse bite, open circuit, short, spur, and spurious copper. In total, it comprises 693 images, which are partitioned into training, validation, and test sets in an 8:1:1 ratio. The image resolution is 3034 × 1586. Representative examples of each defect category are presented in Figure 8.

A self-built chip surface defect dataset was manually labeled using the LabelImg annotation tool, and the dataset consists of five categories: Tre_Etching, Pit, Over_Corrosion, CT_Etching, and Terminal_Error. As shown in Figure 9a, Tre_Etching typically exhibits an irregular shape, is bright white in color, and features distinct black lines. As shown in Figure 9b, the common characteristic of pits is their circular shape, which is further defined by three features: (1) the center is black, surrounded by a light-colored edge forming a circular spot, and (2) the defect appears as a shallow, light-colored circular area. As shown in Figure 9c, Over_Corrosion is usually circular, with a distinct black center. As depicted in Figure 9d, CT_Etching typically has two features: (1) an irregular shape that matches the surface color, with the black line obscured, and (2) an elliptical or circular shape with a black edge in the center, with the black line obscured. As shown in Figure 9e, Terminal_Error is typically located in the terminal area and affects the real defects of the terminal area’s film layer. The specific quantity of each defect category is shown in Table 1. Additionally, the dataset was augmented using techniques such as flipping, translation, and brightness adjustment, resulting in an increase in the total number of images from the initial 584 to 1594. The dataset was then divided into training, validation, and test sets in an 8:1:1 ratio. The image resolution is 300 × 300. The experiment utilizing our self-constructed chip surface defect dataset aims to validate the performance of our proposed method in more challenging real-world scenarios. Through a comparative analysis with a diverse dataset, we seek to demonstrate the versatility and effectiveness of our approach.

### 4.2. Experimental Setup

The experimental platform uses an Intel(R) Core(TM) i7-14700KF CPU and an NVIDIA GeForce RTX 4090 GPU. This platform is capable of handling most deep learning tasks and is suitable for models of varying complexity. The system is equipped with 32 GB of RAM, which meets the requirements for training. The operating system of the experimental platform is Windows 11, and the Python compiler version is Python 3.9.19. The deep learning framework is PyTorch 1.12.0, and the CUDA version is 12.7. Table 2 provides a comprehensive breakdown of the configuration of the experimental conditions. The training parameters are configured as follows: the learning rate is set to 0.01, with stochastic gradient descent (SGD) as the optimizer. The training batch size is 16, and the model is trained for a total of 100 epochs.

To assess the model’s performance in detecting chip surface defects, this study utilizes several evaluation metrics, including precision, recall, mean average precision (mAP), floating-point operations (FLOPs), and model parameters.

Mean average precision is commonly used to evaluate model performance in object detection, as shown in the following formula:(12)$$\mathrm{mAP}=\frac{\sum_{i=0}^{n}\mathrm{AP}\left(i\right)}{n},$$(13)$$\mathrm{AP}=\int_{0}^{1}p\left(R\right)\mathrm{dR}.$$

In this context, n represents the number of image categories, with this study setting n = 5. i denotes the number of detection attempts, and AP refers to the average precision for a single category. This study employs mAP@0.5, setting the intersection over union (IoU) threshold to 0.5, to calculate the AP for each category of images and then takes the average across all categories. The mAP@0.5:0.95 indicates the mean average precision (mAP) calculated over IoU thresholds ranging from 0.5 to 0.95.

Precision indicates the accuracy of the model’s detection, representing the proportion of true positive samples among those classified as positive. Recall indicates the model’s detection capability regarding positive samples, representing the proportion of correctly identified positives among all actual positive cases. The formulas for both metrics are as follows:(14)$$\mathrm{Precision}=\frac{\mathrm{TP}}{\mathrm{TP}+\mathrm{FP}},$$(15)$$\mathrm{Recall}=\frac{\mathrm{TP}}{\mathrm{TP}+\mathrm{FN}}.$$

In this context, TP (true positive) refers to the instances where the model successfully predicts positive cases, FP (false positive) indicates the instances where negative cases are incorrectly classified as positive by the model, and FN (false negative) refers to positive cases that the model incorrectly classifies as negative.

FLOPs (floating point operations) represent the computational complexity of the model, indicating the number of floating-point operations involved during the execution of the model, measured in gigaflops (G).

### 4.3. Experimental Results and Analysis

#### 4.3.1. Analysis and Comparison of Different Attention Mechanisms

To evaluate the impact of attention mechanisms on the model’s detection performance, this study presents a comparative analysis of four attention mechanisms, SE [36], CPCA [37], DAttention [38], and SimAM, across two datasets. Data1 is a PCB defect detection dataset, and Data2 is a self-constructed chip surface defect dataset. In the experiments, the attention mechanisms are incorporated above the SPPF module. The experimental results are presented in Table 3.

From the analysis of the results, it is evident that the four attention mechanisms had minimal impact on the model’s parameter size across both datasets. When compared to the YOLOv8 baseline model and the other three attention mechanisms, the SimAM attention mechanism achieved the highest mean average precision (mAP), with accuracy rates of 93.9% and 89.9%, respectively. Although the DAttention mechanism demonstrated higher accuracy, its mAP values were 86.2% and 90.5%. After incorporating the SE module, the model’s mAP value decreased on Data2 to 87.5%, and its accuracy became the lowest among all four attention mechanisms. In contrast, the CPCA module improved mAP values by 3.3% and 1.4% when compared to the YOLOv8 baseline model. These experiments highlight that the inclusion of the SimAM attention mechanism enhances the network’s accuracy and provides a distinct advantage in defect detection.

#### 4.3.2. Ablation Experiments

To assess the impact of the C2f_RVB module, the SimAM attention mechanism, and the TADDH dynamic detection head on model performance in this study, ablation experiments were conducted using two datasets. Data1 is a PCB defect detection dataset, while Data2 is a custom-built dataset for chip surface defects. The results are presented in Table 4.

From the data presented in the table, it is evident that the integration of different modules significantly enhances detection performance. Firstly, with the inclusion of the C2f_RVB module, the number of parameters is reduced to 2.28 M, while mAP@0.5% increases by 1.2% and 2.3%, respectively. This suggests that the module optimizes the feature fusion structure, effectively reducing the model size while maintaining high detection accuracy. The improvement in detection performance with the SimAM module is comparable to that of the C2f_RVB module, with mAP@0.5% increasing by 4.6% and 2.6%. This indicates that directly computing the three-dimensional attention weights can enhance the model’s detection performance with minimal impact on the model size. When the TADDH dynamic detection head is added, the parameter count decreases to 2.24 M, and mAP@0.5% increases by 6.4% and 3.7%, respectively. This demonstrates that the dynamic detection head significantly reduces the number of parameters, making the model more lightweight. It achieves robust detection results by dynamically selecting features through shared convolutions, particularly when target scales are inconsistent due to feature deficiencies. Finally, compared to the YOLOv8 baseline model, RST-YOLOv8 not only reduces the number of parameters by 1.33M but also increases mAP@0.5% to 92.8% and 94.6%, respectively. This significantly improves overall detection performance, with substantial gains in both precision and recall. The improved model provides excellent detection results for small defects, feature-similar defects, and defects of varying sizes.

#### 4.3.3. Comparison of Experimental Results of the PCB Datasets

To validate the performance of the improved model, a comparative analysis was conducted using the RST-YOLOv8 model alongside classic two-stage detection algorithms, such as Faster R-CNN, as well as single-stage detection algorithms, including SSD and other variants of the YOLO series, along with other relevant improved algorithms, on a PCB defect detection dataset. The results are presented in Table 5.

The data presented in the table clearly indicate that the RST-YOLOv8 model achieves an mAP@0.5% of 92.8%, significantly outperforming other models. Additionally, the model contains only 1.7 million parameters and has a computational cost of 6.6 GFLOPS, demonstrating remarkable efficiency. In contrast, while YOLOv10 and YOLOv8 achieve mAP@0.5% values of 84.2% and 82.5%, respectively, they come with a relatively higher number of parameters and computational costs. The most recent YOLOv11 offers lower parameter counts and computational costs, but its detection accuracy (87.2%) remains inferior to that of the RST-YOLOv8 model. Moreover, the SSD and YOLOv5 models strike a balance between detection accuracy and computational cost, yet their performance still falls short of that of the RST-YOLOv8. Although faster-RCNN performs well in certain aspects, its higher parameter count and computational demands result in decreased efficiency. Compared to the findings in the literature [28], the improved algorithm presented in this paper demonstrates superior performance in both detection accuracy and model lightweighting, achieving the highest level of average precision. Overall, the RST-YOLOv8 model not only achieves superior detection accuracy while satisfying real-time detection requirements but also demonstrates exceptional performance in PCB defect detection, highlighting its considerable application value and efficiency.

Figure 10 presents a comparison of the detection performance of the RST-YOLOv8 model with that of other algorithmic models in the identification of PCB defects. The analysis of the detection results reveals that, as shown in Figure 10g, the RST-YOLOv8 model not only exhibits an improvement in the confidence levels of the predicted bounding boxes but also rectifies instances of false positives and missed detections observed in other models. Moreover, the model’s enhanced capability to process features of small-sized defects has led to a significant improvement in the detection performance for spur defects. In conclusion, the RST-YOLOv8 model demonstrates strong recognition performance for PCB defects.

The PR curves presented in Figure 11 and Figure 12 illustrate the experimental results of YOLOv8 and the enhanced RST-YOLOv8 on the PCB dataset. These figures display the mAP@0.5 values for each individual category, as well as the overall mAP@0.5. As shown in the charts, the improved algorithm increases the mAP from 82.5% to 92.8%, yielding an improvement of 10.3 percentage points. Notably, in YOLOv8 detection, the mAP for the “mouse_bite” category is only 0.808, whereas the enhanced model achieves a mAP of 0.941. This highlights a substantial performance gain over the baseline model for this particular category.

#### 4.3.4. Comparison of Experimental Results of the Chip Surface Defect Dataset

To evaluate the performance of the improved model, a comparative analysis was conducted between the RST-YOLOv8 model, the classic two-stage detection algorithm faster-RCNN, the classic single-stage detection algorithm SSD, and other algorithms in the YOLO series, using a self-constructed chip surface defect dataset. The results are presented in Table 6.

The data presented in the table clearly indicate that the RST-YOLOv8 model achieves an mAP@0.5% value of 94.6%, significantly outperforming other models. Furthermore, this model has only 1.68 million parameters and a computational cost of 6.6 GFLOPS, demonstrating exceptional efficiency. In comparison, YOLOv10 and YOLOv8 achieve mAP@0.5% values of 91.6% and 89.2%, respectively, but with higher parameter counts and computational costs. Although the latest YOLOv11 features reduced parameters and computational costs, its detection accuracy (88.2%) remains lower than that of the RST-YOLOv8 model. Additionally, while the SSD and YOLOv5 models strike a relatively balanced trade-off between detection accuracy and computational cost, their overall performance lags behind that of the RST-YOLOv8. In conclusion, the RST-YOLOv8 model not only satisfies real-time detection requirements but also achieves the highest detection accuracy, demonstrating its superior effectiveness in chip surface defect detection.

The PR curves presented in Figure 13 and Figure 14 illustrate the experimental results of YOLOv8 and the enhanced RST-YOLOv8 on a custom-built chip surface dataset. These figures display the mAP@0.5 values for each category, as well as the overall mAP@0.5. From the graphs, it is evident that the improved algorithm increased the mAP from 89.2% to 94.6%, yielding a 5.4 percentage point improvement. Notably, in YOLOv8 detection, the mAP value for the “Terminal_Error” category was only 0.662, while the improved model achieved 0.804. This demonstrates a significant enhancement relative to the baseline model for this specific category.

Figure 15 illustrates a comparison of defect detection performance between the RST-YOLOv8 model and other algorithmic models for identifying surface defects on chips. As shown in Figure 15g, the RST-YOLOv8 model exhibits increased sensitivity to small defects. The original model encountered false positives and missed detections on the chip surface, but these issues were addressed through the improvements. Furthermore, by integrating feature information from defects of varying sizes, the detection accuracy for CT_Etching defects has been notably enhanced. In conclusion, the RST-YOLOv8 model demonstrates superior defect recognition capability for chip surfaces.

#### 4.3.5. Convergence Analysis

Figure 16 presents the performance evaluation results of the object detection model based on the improved architecture over 100 training epochs, along with an analysis of the loss function and key metrics during both the training and validation phases. The results demonstrate that the box loss, class loss, and distributed focal loss (DFL) all decreased significantly as training progressed, indicating that the model gradually converged and enhanced its accuracy in object recognition and classification tasks. Both precision and recall showed substantial improvements, ultimately stabilizing at a high level, which further validates the model’s efficiency in object detection. Furthermore, the mean average precision (mAP) exhibited consistent improvement across various thresholds, highlighting the model’s strong detection performance even under varying overlap conditions. Overall, the experimental results confirm the model’s effectiveness and robustness for object detection tasks, with the potential for further performance improvements through optimization of the model architecture or an increase in the training data volume.

## 5. Conclusions

This paper addresses the challenge of low detection accuracy in chip defect detection, which arises from the similarity of defect features, the small size of defect targets, and significant variations in defect scales. To tackle this issue, we propose a defect detection method based on an improved YOLOv8 model. Specifically, we design the C2f_RVB module, which leverages RepViTBlock technology to enhance feature representation. This optimization not only reduces the model’s parameter count but also significantly improves the detection accuracy for small defect targets. In addition, we introduce the SimAM attention mechanism, enabling the model to better capture and process information from three-dimensional channels, thereby improving its ability to perceive defect features. Furthermore, we enhance the original detection model by incorporating an alignment dynamic detection head specifically designed to address small target chip defect detection. This detection head facilitates task interaction between the localization and classification branches, leading to more accurate predictions by adjusting the results of both branches. Experimental results demonstrate that the improved model outperforms existing models in terms of accuracy, recall, and mean precision. This indicates significant advantages over other models.

Future research will focus on exploring techniques such as quantization, structured pruning, and knowledge distillation to further reduce model size and computational cost while maintaining high accuracy. Additionally, by optimizing operator fusion and memory access patterns, the model will be tailored for edge devices to enhance real-time performance in industrial deployments. Simultaneously, the model’s applicability will be extended to accommodate various chip types and defect scenarios, ensuring robust performance under diverse manufacturing conditions. These efforts will significantly contribute to enhancing the practicality and efficiency of the RST-YOLOv8 model in real-world chip defect detection applications.

## Figures and Tables

**Figure 1 sensors-25-03859-f001:**
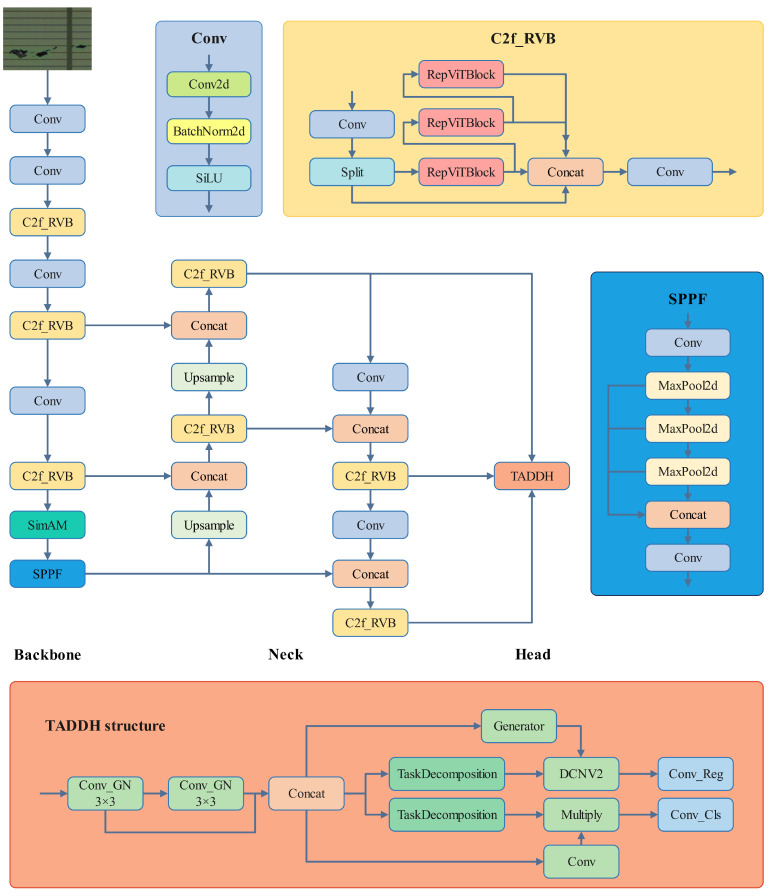
RST-YOLOv8 network architecture.

**Figure 2 sensors-25-03859-f002:**
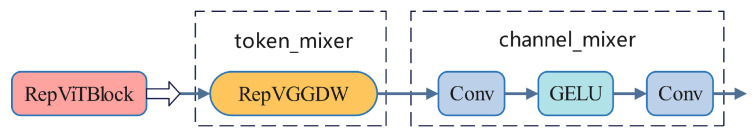
RepViTBlock structure.

**Figure 3 sensors-25-03859-f003:**
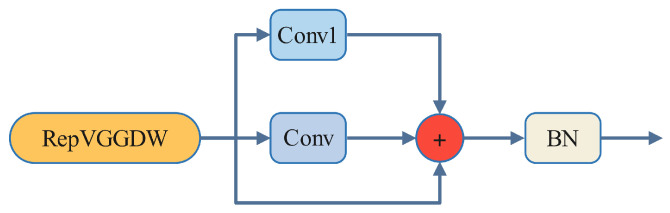
RepVGGDW structure.

**Figure 4 sensors-25-03859-f004:**
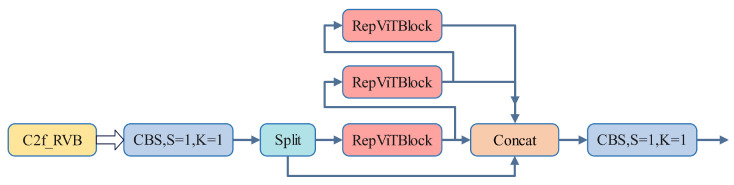
C2f_RVB structure.

**Figure 5 sensors-25-03859-f005:**
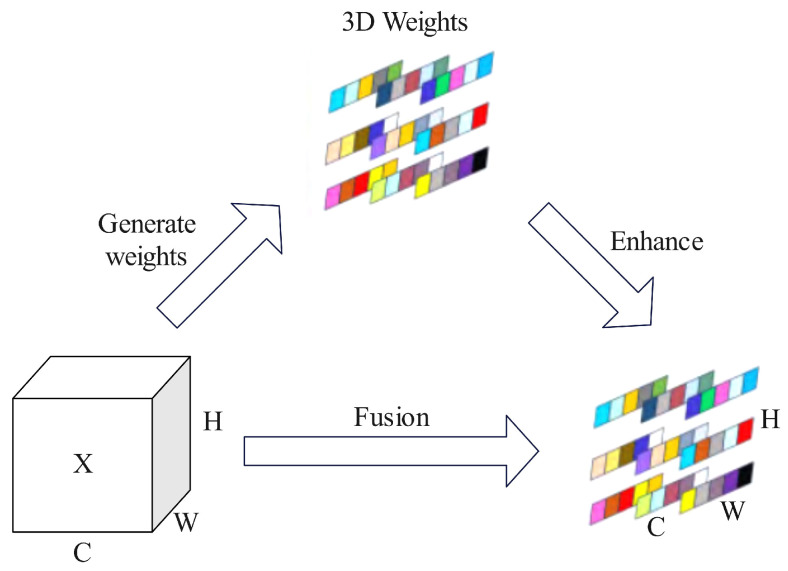
SimAM attention mechanism structure diagram.

**Figure 6 sensors-25-03859-f006:**
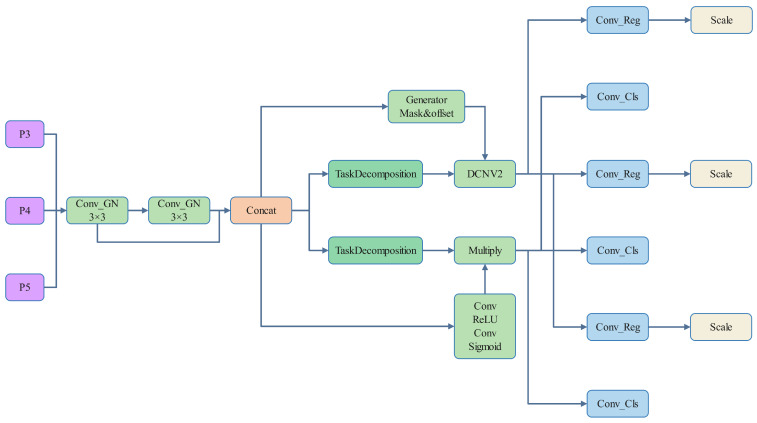
TADDH structure.

**Figure 7 sensors-25-03859-f007:**
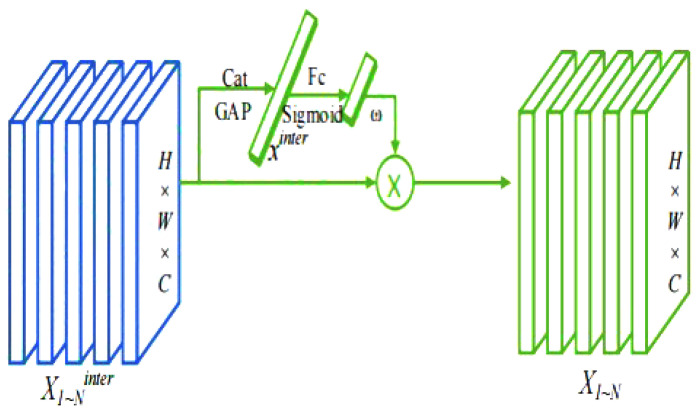
Schematic diagram of task decomposition structure.

**Figure 8 sensors-25-03859-f008:**
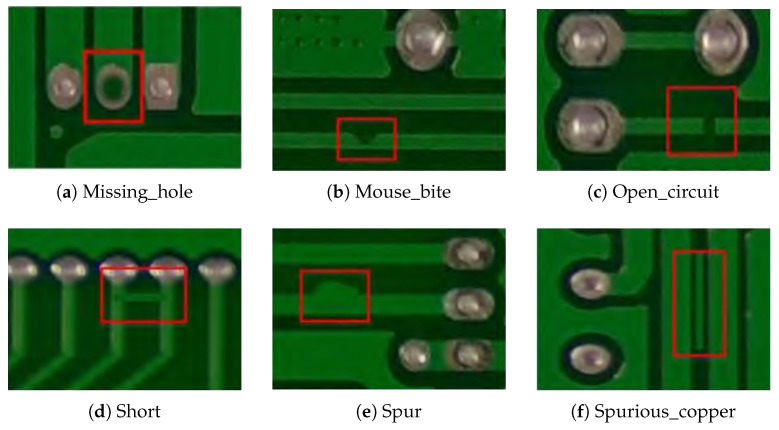
PCB defect sample.

**Figure 9 sensors-25-03859-f009:**
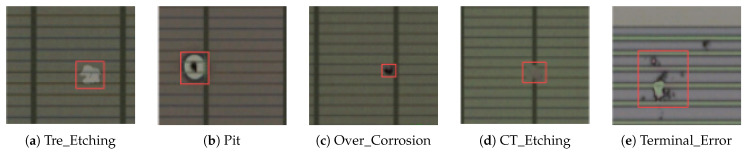
Chip surface defect sample.

**Figure 10 sensors-25-03859-f010:**
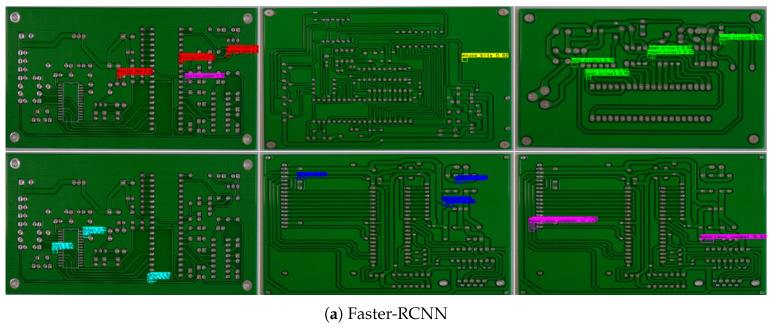
Comparison of PCB dataset detection effects.

**Figure 11 sensors-25-03859-f011:**
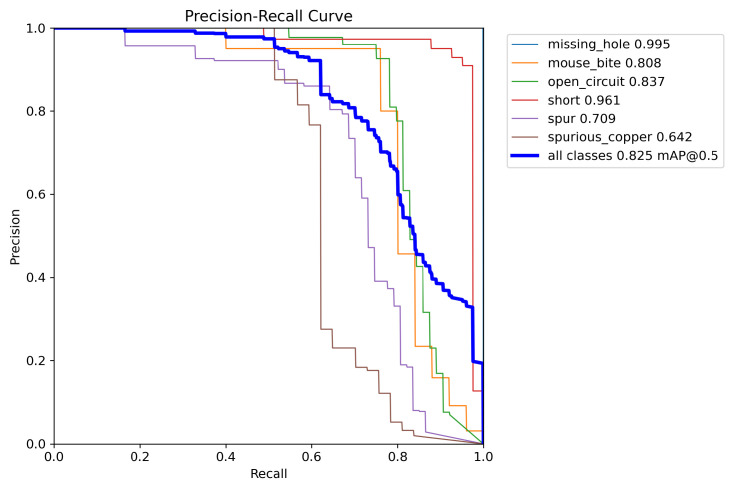
Training results of the YOLOv8 algorithm on PCB dataset.

**Figure 12 sensors-25-03859-f012:**
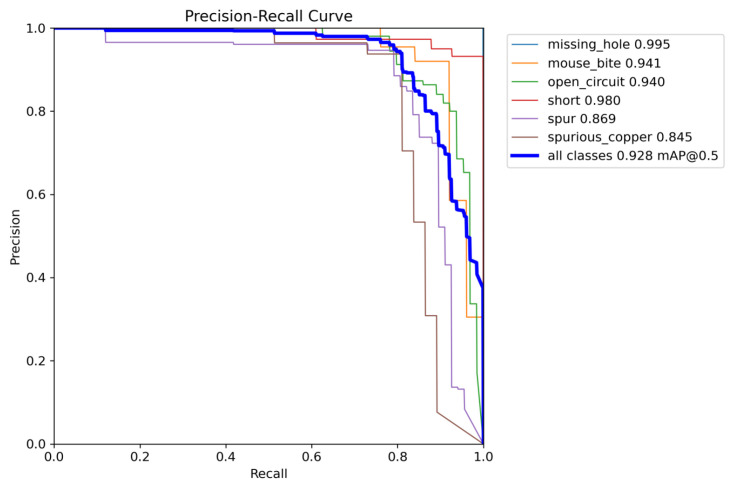
Training results of the RST-YOLOv8 algorithm on PCB dataset.

**Figure 13 sensors-25-03859-f013:**
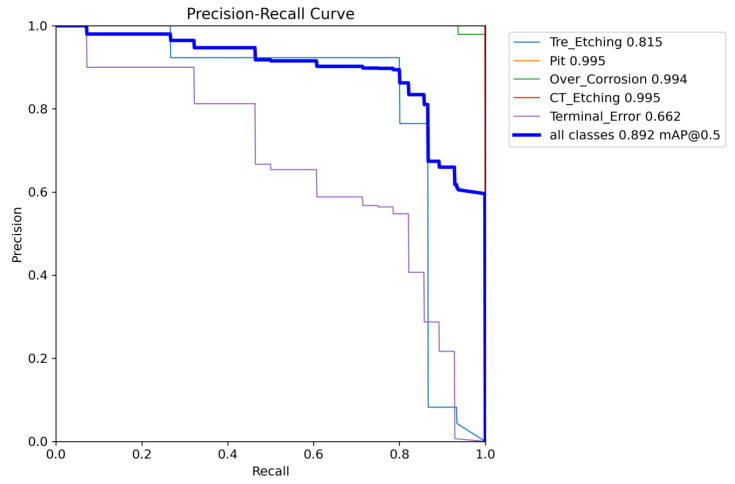
Training results of the YOLOv8 algorithm on Chip Surface Defect Dataset.

**Figure 14 sensors-25-03859-f014:**
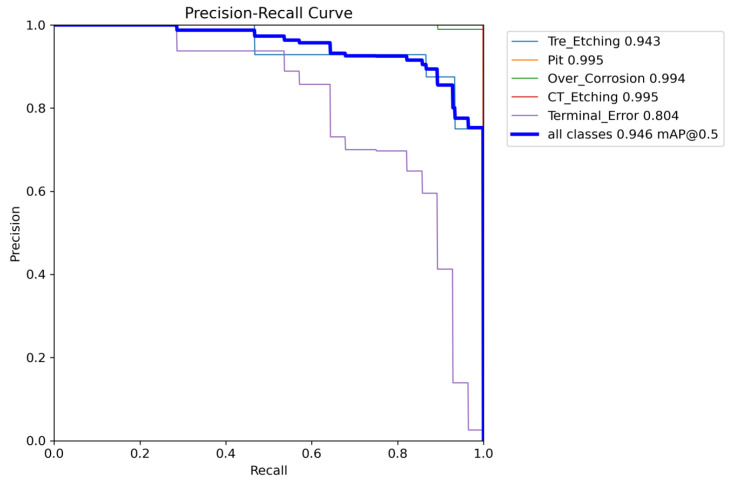
Training results of the RST-YOLOv8 algorithm on Chip Surface Defect Dataset.

**Figure 15 sensors-25-03859-f015:**
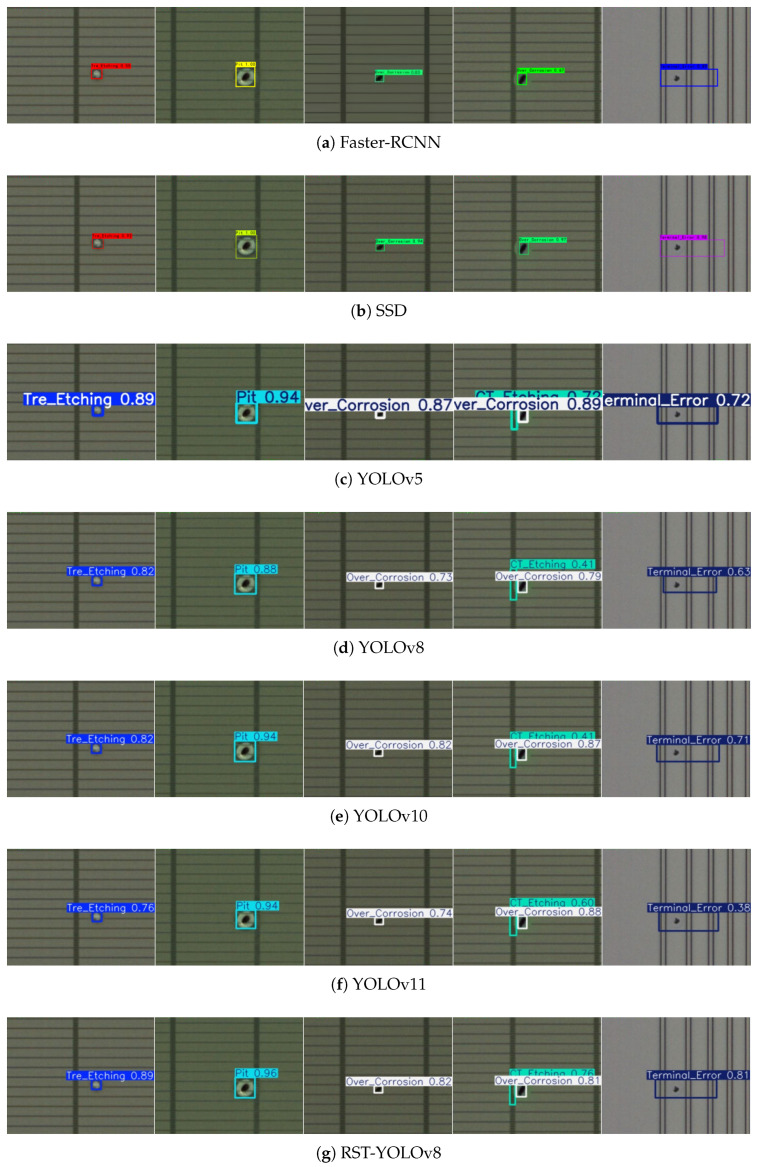
Comparison of chip surface defect detection effects.

**Figure 16 sensors-25-03859-f016:**
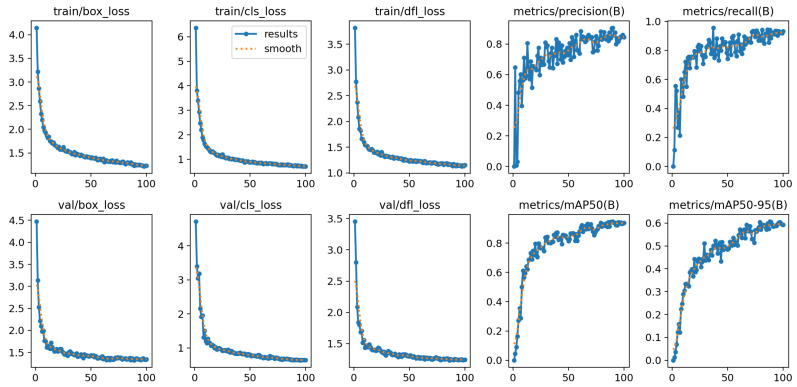
The network convergence of the RST-YOLOv8 algorithm on the Chip Surface Defect Dataset.

**Table 1 sensors-25-03859-t001:** Chip surface defect number distribution table.

Category	Number
Tre_Etching	153
Pit	108
Over_Corrosion	912
CT_Etching	144
Terminal_Error	307

**Table 2 sensors-25-03859-t002:** Configuration of the experimental conditions.

Name	Specific Information
Operating system	Windows11
CPU	Intel(R) Core(TM) i7-14700KF
	@5.60 GHz
GPU	NVIDIA GeForce RTX 4090
RAM	32 GB
CUDA	12.7
PyTorch	1.12.0
Python	3.9.19

**Table 3 sensors-25-03859-t003:** Comparison of different attention mechanisms.

Models	Data1				Data2			
	P/%	R/%	mAP@0.5/%	Para/M	P/%	R/%	mAP@0.5/%	Para/M
YOLOv8n	91.8	76.1	82.5	3.01	70.0	89.3	89.2	3.01
+SE	91.3	79.5	85.9	3.02	79.7	84.5	87.5	3.02
+CPCA	91.0	78.1	85.8	3.13	85.5	85.1	90.6	3.13
+DAttention	94.9	79.4	86.2	3.27	87.5	86.5	90.5	3.27
+SimAM	93.9	75.4	87.1	3.01	89.9	75.4	91.8	3.01

**Table 4 sensors-25-03859-t004:** Ablation experiments.

Methods				Data1				Data2			
YOLOv8	C2f_RVB	SimAM	TADDH	P/%	R/%	mAP@0.5/%	Para/M	P/%	R/%	mAP@0.5/%	Para/M
✓				91.8	76.1	82.5	3.01	70.0	89.3	89.2	3.01
✓	✓			90.1	75.6	83.7	2.28	88.7	84.1	91.5	2.28
✓		✓		93.9	75.4	87.1	3.01	89.9	75.4	91.8	3.01
✓			✓	91.0	84.8	88.9	2.24	77.1	92.3	92.9	2.24
✓	✓	✓		89.2	83.8	88.1	2.28	82.8	88.9	92.9	2.28
✓	✓		✓	89.9	86.4	90.3	1.64	92.5	81.8	93.1	1.64
✓		✓	✓	93.9	83.0	88.4	2.24	83.4	92.8	93.9	2.24
✓	✓	✓	✓	94.4	86.1	92.8	1.68	83.7	94.3	94.6	1.68

**Table 5 sensors-25-03859-t005:** Comparison with other algorithms on PCB dataset.

Models	P/%	R/%	mAP@0.5/%	mAP@0.5:0.95/%	Para/M	GFLOPs
Faster-RCNN	80.5	89.0	74.6	43.2	41.3	205.1
SSD	73.8	72.1	77.7	36.8	26.3	85.7
YOLOv5	79.8	76.1	81.3	56.4	7.0	15.8
YOLOv8	91.8	76.1	82.5	63.1	3.0	8.2
YOLOv10	84.2	76.8	84.2	64.9	2.3	6.5
YOLOv11	93.8	78.5	87.2	70.3	2.6	6.4
Literature [28]	89.7	83.5	87.7	75.3	2.6	-
RST-YOLOv8	94.4	86.1	92.8	78.5	1.7	6.6

**Table 6 sensors-25-03859-t006:** Comparison with other algorithms on Chip Surface Defect Dataset.

Models	P/%	R/%	mAP@0.5/%	mAP@0.5:0.95/%	Para/M	GFLOPs
Faster-RCNN	52.5	82.1	77.7	47.8	41.3	205.1
SSD	92.5	50.8	83.6	53.2	26.3	85.7
YOLOv5	92.1	83.9	86.8	61.4	7.0	15.8
YOLOv8	70.0	89.3	89.2	65.3	3.0	8.2
YOLOv10	78.0	88.7	91.6	70.6	2.3	6.5
YOLOv11	82.5	82.5	88.2	62.9	2.6	6.4
RST-YOLOv8	83.7	94.3	94.6	74.5	1.7	6.6

## Data Availability

Data are contained within the article.

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
