# Peer review of "RST-YOLOv8: An Improved Chip Surface Defect Detection Model Based on YOLOv8"

_sensors, 2025, doi:10.3390/s25133859_

Round 1
Reviewer 1 Report
Comments and Suggestions for Authors
The proposed architectural improvements to YOLOv8, particularly the integration of the C2f_RVB module, SimAM attention, and the Task-Aligned Dynamic Detection Head (TADDH), present an interesting method for enhancing defect detection methodologies. However, to more effectively highlight the paper's contributions and situate it within the existing body of research, several areas could be significantly strengthened.
A key area for improvement lies in the depth and specificity of the related work. While the paper aims to tackle challenges associated with detecting small defects, the current discussion does not sufficiently engage with the extensive literature dedicated to small object detection (SOD). A more comprehensive review would explore various established SOD strategies, such as advanced feature pyramid designs, specialized data augmentation techniques, or context-aware methodologies, and discuss how RST-YOLOv8's components relate to or differentiate from these. Furthermore, a more focused analysis of prior research that has specifically utilized the same public Peking University PCB defect dataset is essential. Without a thorough review of how other researchers have approached this dataset and their reported outcomes, it becomes difficult to accurately gauge the advancements offered by RST-YOLOv8 in this particular context.
This leads to a significant concern regarding comparative analysis and benchmarking. The paper currently lacks a direct comparison of RST-YOLOv8's performance against other published models on the Peking University PCB dataset. A literature search reveals several other studies that have employed this dataset, and it is crucial to benchmark RST-YOLOv8 against these existing results. Had such a comparison been undertaken, it is possible that other models, perhaps those leveraging extensive data augmentation or different architectural choices, might exhibit comparable or even superior performance metrics on this specific public benchmark. This omission makes it challenging to definitively assess the performance gains achieved by RST-YOLOv8 relative to the current state-of-the-art for this dataset.
The absence of these direct comparisons and a broader contextualization within specialized literature directly impacts the framing of the paper's novelty and contribution. If other methodologies have already demonstrated robust performance on the same public dataset for PCB defect detection, claims regarding RST-YOLOv8 significantly advancing the field for this task need careful consideration. While the architectural integrations of C2f_RVB, SimAM, and TADDH into the YOLOv8 framework are certainly noteworthy, and the ablation studies effectively demonstrate their individual benefits, the overall contribution of the paper should be articulated with greater precision. Instead of focusing on general goals like "addressing the challenges of detecting small defects in low-resolution images," which may have been substantially tackled by prior art with potentially better results on some benchmarks, the paper could more strongly emphasize the specific advancements it brings. This might include highlighting the unique architectural synergy of the proposed components within YOLOv8, the achieved balance between performance and computational efficiency (especially considering the parameter reduction), or its performance characteristics perhaps under conditions of limited data augmentation compared to models heavily reliant on it, particularly when discussing the results from both the public PCB dataset and the self-built chip defect dataset.
To sum it up, I strongly encourage the authors to:
+ Expand related work to thoroughly cover small object detection techniques and prior studies on surface defect detection.
+ Incorporate direct comparative results against other models on the Peking University PCB dataset, along with a nuanced discussion of these comparisons.
+ Refine the claims of novelty and contribution to focus on the specific advancements to the YOLOv8 architecture and its resulting performance profile.
Reviewer 2 Report
Comments and Suggestions for Authors
The paper presents a clear research objective and a logically structured methodology. However, several detailed revisions are recommended to enhance clarity and academic rigor:
- The abstract currently reports multiple detection accuracy metrics (mAP@0.5, mAP@0.5:0.95, and recall) to highlight performance improvements. To improve clarity, the authors should simplify the presentation and retain only one representative mAP metric (e.g., mAP@0.5) for conciseness. Additionally, since Table 5 and 6 in the experiments compare computational complexity (e.g., parameters and GFLOPs), the abstract should explicitly mention these metrics to emphasize the model’s strengths in both high accuracy and lightweight design.
- The keywords currently used are not accurate. Since the improvement module is not proposed by this study, it is recommended that the keywords C2f_RVB; SimAM; TADDH be deleted and words that reflect the content of the study be added.
- The Introduction lacks a brief organization of the paper. A paragraph summarizing the structure (e.g., "Section 2 reviews YOLOv8") would help readers navigate the manuscript more effectively.
- With regard to replaced or added modules, streamlining their rationale requires an explanation of their specific role in detection, in the context of the objective being detected.
- The SimAM attention mechanism (Section 3.4) was originally proposed by Yang et al. in 2021 (SimAM: A Simple, Parameter-Free Attention Module for Convolutional Neural Networks, ICML 2021). The authors must cite this work correctly in the methodology.
- The PKU-Market-PCB dataset (Data1) requires formal citation in Section 4.1.
- The authors mention FPS as an evaluation metric for model detection speed in the experimental setup in section 4.2, but it is not listed in the table of experimental results.
- The defect detection results in Figures 10 and 15 are presented at small scales, making it difficult to compare improvements. To enhance visual interpretability, the authors should add zoomed-in subfigures for defects to clearly showcase detection quality.
- The current comparative experiments (Figure 10 and 15) display results from different models in separate subfigures. To improve clarity, the authors should adopt side-by-side comparisons of the same image across models for direct evaluation of detection performance.
Round 2
Reviewer 1 Report
Comments and Suggestions for Authors
The authors addressed all of my comments.